# Substituent Effects in the Thermal Decomposition of 1,2,4-Triazol-3(2H)-Ones and Their Thione Analogues: A DFT Study with Functional Performance

**DOI:** 10.3390/molecules31010109

**Published:** 2025-12-27

**Authors:** Rosalinda Ipanaque-Chávez, Marcos Loroño, Tania Cordova-Sintjago, José L. Paz

**Affiliations:** 1Departamento Académico de Fisicoquímica, Facultad de Química e Ingeniería Química, Universidad Nacional Mayor de San Marcos, Lima 15081, Peru; rosalinda.ipanaque@unmsm.edu.pe; 2Department of Natural Sciences, Santa Fe College, Gainesville, FL 32066, USA; tania.cordova-sintjago@sfcollege.edu; 3Departamento Académico de Inorgánica, Facultad de Química e Ingeniería Química, Universidad Nacional Mayor de San Marcos, Lima 15081, Peru; jpazr@unmsm.edu.pe

**Keywords:** DFT, triazole, regioselective, kinetics, reaction mechanisms, NBO, IGM

## Abstract

This computational study investigates the thermal decomposition of 1,2,4-triazol-3(2H)-ones and their thione analogues using Density Functional Theory (DFT). The reaction proceeds via a concerted, six-membered cyclic transition state, primarily driven by the breaking of the N–N bond. A key finding is that the accuracy of the calculated activation energies (Ea) strongly depends on the choice of DFT functional. For sulfur-containing systems (thiones), the hybrid functional APFD (with 25% Hartree–Fock exchange) provides the most reliable results, effectively describing their higher polarizability. In contrast, for oxygen-containing systems (triazolones), the dispersion-corrected functional B97D-GD3BJ (with 0% Hartree–Fock exchange) delivers superior accuracy by better modeling electrostatic and dispersion interactions. The -CH_2_CH_2_CN group at the N-2 position acts not only as a protecting group but also stabilizes the transition state through non-covalent interactions. Electron-withdrawing substituents slightly increase the E_a_, while electron-donating groups decrease it. Sulfur analogues consistently show significantly lower activation energies (by ~40 kJ/mol) than their oxygen counterparts, explaining their experimentally observed faster decomposition. This work establishes a dual-methodology computational framework for accurately predicting the kinetics of these reactions, providing valuable insights for the regioselective synthesis of biologically relevant triazole derivatives via controlled pyrolysis.

## 1. Introduction

Nitrogen-containing five-membered heterocycles, particularly triazoles, play a pivotal role in medicinal and materials chemistry due to their pharmacological versatility and thermal stability [1,2]. Among these, 1,2,4-triazole derivatives stand out for their broad biological activities, including antifungal, antiviral, and anticancer properties [3,4,5,6,7]. 1H-1,2,4-triazole-5(4H)-one is depicted in Figure 1b. Substituting oxygen for sulfur to form triazole-thione derivatives enhances its antiproliferative and anticonvulsant effects, as documented in previous studies [8,9] (Figure 1).

An experimental work by Al-Awadi et al. studied the thermal decomposition of 4-arylideneimino-2-cyanoethyl-1,2,4-triazole-3(2H)-ones/thiones in the gas phase and proposed a unimolecular concerted mechanism via a six-membered cyclic transition state (TS), as shown in Figure 2. Triazole-thione derivatives exhibited reactivity 10^3^ times faster than their oxygenated analogues at 500 K [10]. While this difference in reactivity was initially attributed to the greater thermodynamic stability of π bonds (C=S vs. C=O) and enhanced protophilicity [10,11], the underlying electronic and steric details remained unresolved.

This study focuses on 2-ethyl or 2-(2-cyanoethyl)-1,2,4-triazol-3(4H)-ones/thiones (R = CH_2_CH_2_CN) with N-2 substitution, aiming to facilitate regioselective functionalization at other ring nitrogen sites. Controlled pyrolysis of 2-cyanoethyl derivatives allows sequential deprotection at N-4 and N-2, offering a strategic pathway for synthesizing 2- and 4-substituted 1,2,4-triazoles with potential biological applications. For the studied system, Al-Awadi et al. highlighted the critical influence of the -CH_2_CH_2_CN group at the triazole N-2 position on activation energy (*E_a_*) through steric and electronic interactions with the TS [12].

In this theoretical work, we are re-evaluating the oxygen/sulfur influence in the reaction mechanism and assessing the sulfur atom effect on the E_a_ and exploring the intrinsic electronic dynamics, including the role of hybrid functionals in Hartree–Fock exchange (%HF) contributions and their impact on E*_a_* predictions. We employ the APFD (Austin–Frisch–Petersson with dispersion) method [13], paired with the Def2-TZVP basis set, as a reference, alongside other functionals with varying percentages of HF contributions. Non-covalent interactions in the TS, involving compounds other than -CH_2_CH_2_CN, such as: -H; -CH_2_CH_3_; -CH_2_CH_2_-CS-CH_3_; -CH_2_CH_2_-CO-CH_3_; and -CH_2_CH_2_-S-CH_3_, were also studied to understand their possible influence on the reaction mechanism. The different TSs were analysed via the Independent Gradient Model (IGM) [14,15,16], through the descriptors δ*g* or δ*q*, to elucidate reactivity differences between S/O analogues.

## 2. Results and Discussion

The mechanism depicted in Figure 2 was used for theoretical calculations, using a variety of functionals. The cyclic six-membered transition state in these eliminations involves a proton transfer to oxygen or sulfur, resulting in the formation of the enol form of the triazolone or triazothione, along with an aromatic nitrile, as illustrated in Figure 3. The enol form tautomerizes with the corresponding keto form of the product, which is the product isolated in experimental work.

To ensure a clear and logical flow, this work is organized as follows. The next section details the selection and justification of the six-membered cyclic transition state model for the decomposition reaction. The analysis is then developed across seven dedicated subsections, spanning from the foundational computational setup to the application of the Independent Gradient Model (IGM) for topological analysis. This structure provides a comprehensive pathway to evaluate the accuracy and mechanistic insights of our theoretical approach.

### 2.1. Model Selection

To determine the most appropriate model, we compared the calculated thermodynamic parameters with the experimental values. Table 1 shows the thermodynamic parameters and activation energies (Ea) calculated at APFD/Def2-TZVP (25% HF) theoretical level, for 4-arylideneimino-1,2,4-triazol-3(2H)-ones (I–V) and their thione analogues (VI–X), shown in Figure 3, with R = H. For sulfur derivatives (VI–X), discrepancies emerge between calculated and experimental Ea values, particularly significant for Y = H (29.0% error) and Y = Cl (27.1% error). Oxygen analogues (I–V) show consistently high deviation (average ~20%), peaking at 39.0% for Y= -OMe. Calculations reveal a systematic ~40 kJ/mol gap between sulfur (X = S) and oxygen (X = O) derivatives, favoring decomposition in sulfur systems (160.40 kJ/mol for X = S/Y = H vs. 199.77 kJ/mol for X = O/Y = H). However, critical inconsistencies persist: the X = O/Y = -OMe case shows the highest discrepancy (39.0%, Ea = 197.47 kJ/mol vs. exp. 142.1 kJ/mol), while for sulfur, substituents Y = H and Cl yield unusually high errors (29.0% and 27.1%), despite the exceptional accuracy for X = S/Y = -OMe (1.4% error). These divergences underscore the APFD functional’s sensitivity to combining electronegativity, steric, and polarity effects of substituents.

The pronounced 29.0% error in activation energy (Ea) for sulfur (X = S/Y = H) with APFD (25% HF), contrasted with 14.4% for oxygen (X = O/Y = H), stems from how Hartree–Fock (HF) exchange modulates self-interaction error (SIE) in transition states (TS). Sulfur’s high polarizability exacerbates SIE in hybrid functionals: while pure/near-zero HF functionals like B97D-GD3BJ (0% HF) yield exceptional accuracy for X = S/Y = H (Ea = 121.59 kJ/mol, −2.2% error), hybrids with >20% HF (e.g., wB97XD at 100% HF) overestimate Ea by 40.7–55.2% due to exaggerated charge localization. APFD’s 25% HF strikes a poor balance for sulfur, over-stabilizing reactants by inadequately correcting delocalization in the TS.

To gain a better understanding of the errors in the determinations of activation energies, several functionals were selected with varying percentages of Hartree–Fock (HF), ranging from 0% to 100%. The results are presented in Table 2 and Table 3 for 4-benzylideneimino-1,2,4-triazole-3(2H)-one (2H) and 4-benzylideneimino-1,2,4-triazole-3(2H)- one (R = H, Y = H), as displayed in bold characters in Table 1. Oxygen-containing TSs tolerate moderate HF exchange: B3LYP-GD3BJ (20% HF) reduces the error to 5.8% for X = O/Y = H, while pure functionals as BLYP (0% HF) underestimate Ea by 13.8% due to insufficient exchange repulsion. APFD’s intermediate 25% HF performs better for oxygen (14.4% error) than sulfur, where low polarizability reduces SIE sensitivity. Sulfur’s diffuse electrons demand minimal HF exchange (0–15%) to model charge dispersion during N–N bond rupture, whereas oxygen’s lesser polarizability benefits from 20 to 25% HF to avoid spurious stabilization.

This dichotomy explains APFD’s failure for X = S/Y = H and highlights the need for system-specific HF tuning: sulfur requires dispersion-corrected low-HF functionals (e.g., B97D), while oxygen achieves optimal accuracy with moderate-HF hybrids (e.g., B3LYP, or B97D).

As shown, the systematic benchmarking in Table 2 and Table 3 reveals that APFD (25% HF) provides the most reliable activation energies for sulfur-containing triazoles, maintaining consistent errors (10–13% for X = S/R = H/Y = substituents) while accurately capturing substituent trends despite a slight overestimation versus pure functionals. Conversely, for oxygen analogues, B97D-GD3BJ (0% HF) delivers superior accuracy, reducing errors to ≤7.5% (Table 3: X = O/Y = H E_a_ = 160.62 kJ/mol vs. exp. 174.6 kJ/mol, −6.7% error) by mitigating self-interaction error and better describing electrostatic/dispersion forces in the transition state. This dichotomy, where sulfur’s polarizability favours APFD’s hybrid character, and oxygen’s lesser polarizability benefits from B97D’s dispersion-corrected pure DFT, establishes the dual-methodology approach for studying cyano group effects (Figure 1).

DFT calculations have been used to explain the differences [18] in reactivities of oxygen and sulfur-containing species, in terms of the hardness and softness of acids and bases, HSAB. Calculations show the frontier orbitals of sulfur compounds have a smaller energy gap compared to their oxygen analogs, explaining the increased orbital interaction (covalency) in sulfur-based bonding. DFT-based population analyses, like Mulliken or Bader charges, reveal the electrostatic character of bonds. Hard-hard interactions (like those with oxygen) are dominated by charge-charge attraction, while soft-soft interactions (like those with sulfur) are driven by orbital overlap.

### 2.2. Application to 4-Arylideneimino-R-1,2,4-triazol-3(2H)-ones/thiones

Having identified a suitable model for the study, we proceeded to vary the R group from H to -CH_2_-CH_2_-CN and the Y group to H, NO_2_, Me, OMe, and Cl, in 4-arylideneimino-1,2,4-triazol-3(2H)-one and thione analogues, Figure 1. Activation energies were calculated at the APFD-Def2-TZVP level of theory to verify the model already established.

Table 4 shows the calculated thermodynamic parameters for the structures shown in Figure 2, using B97D-GD3BJ for 4-arylideneimino-1,2,4-triazol-3(2H)-one derivatives and APFD for the corresponding thiones; all calculations were carried out with Def2-TZVP basis set. The transition state has a six-member ring geometry, as shown in Figure 1.

Table 5 shows the results obtained for the case of the functional B97D-GD3BJ. The table also indicates other R groups, including: CH_2_CH_2_-CO-CH_3_, CH_2_CH_2_-CS-CH_3_, CH_2_CH_2_-O-CH_3_, and CH_2_CH_3_. These groups were aggregated to affect the six-ring transition state mechanism. In general, the activation energy remains constant at about 160 kJ/mol.

Table 5 also shows that the O-compounds (R = CH_2_CH_2_CN, Y = H) at the B97D-GD3BJ theoretical level; the calculated activation energy was 162.21 kJ/mol, aligning closely with experimental values (error: −6.7%), whereas APFD overestimates (Ea = 199.50 kJ/mol, +23.4% error). Conversely, for S-systems (R = CH_2_CH_2_CN, Y = Cl), APFD predicts Ea = 163.36 kJ/mol with minimal error (0.2–13.1%), while B97D-GD3BJ underestimates (Ea = 124.68 kJ/mol). This divergence stems from B97D-GD3BJ’s superior handling of dispersion-driven non-covalent interactions in O-TS (0% HF exchange), in contrast to APFD’s 25% HF exchange better captures S-TS steric/electronic effects during N–N bond cleavage.

We found that substituent effects on energies of activation are small for O-systems and S-systems: electron-withdrawing groups (NO_2_) raise Ea for both systems (O: 162.21 → 166.14 kJ/mol; S: 162.34 → 166.67 kJ/mol), while electron-donating groups (OCH_3_/CH_3_) slightly reduce Ea (S-OCH_3_: 162.03 kJ/mol). S-compounds consistently exhibit lower Ea than O-analogues (ΔEa ≈ 37 kJ/mol), confirming to be in agreement with experimental values; 10^3^× faster thiocarbonyl decomposition. Hybrid functionals with high HF exchange (M062X, 54% HF) overestimate Ea by 43–55% for S-systems, underscoring APFD’s balanced parametrization. For regioselective synthesis, B97D-GD3BJ/Def2-TZVP is recommended for O-triazoles, while APFD/Def2-TZVP optimizes predictions for S-triazolothiones, enabling precise kinetic control [19].

The experimental work by Al-Awadi et al. also demonstrated that the -CH_2_CH_2_CN group on R undergoes pyrolytic decomposition at 800 K, as shown in Figure 4. Our DFT calculations at the APFD/Def2-TZVP level explored this side reaction by replacing the -CN group with -H, -CS-CH_3_, -S-CH_3_, and -CO-CH_3_. The results indicate that these alternative decompositions occur at higher activation energies, consistently above 200 kJ/mol, making them less favorable than the primary six-membered ring mechanism. For the specific case of the -CH_2_CH_2_-CS-CH_3_ group, reported in Table 6, the calculated activation energies are notably comparable to those of the main pathway. For instance, with sulfur (X = S), APFD predicts an E_a_ of 149.75 kJ/mol, while B97D-GD3BJ gives a lower value of 107.80 kJ/mol. This suggests that this side reaction could compete with the main reaction pathway for sulfur-containing systems where the energy barrier is significantly lower.

### 2.3. Non-Covalent Interactions (NCI)

In this work, we looked at the weak forces, called non-covalent interactions (NCIs), that come from the -CH_2_CH_2_CN group in our reaction. We found that this group does more than just act as a protective group. It also uses these weak forces to help stabilise the transition state—the key step in the reaction. These NCIs are not strong like a chemical bond but together they have a big impact. They help control where and how fast the reaction happens (the regioselectivity and kinetics). For the sulfur-containing molecules, the -CH_2_CH_2_CN group positions itself in a way that creates helpful attractions with the sulfur atom. This extra stabilization makes the reaction happen much more easily, lowering the energy needed by about 40 kJ/mol. This is why the sulfur versions react about 1000 times faster than the oxygen ones. For the oxygen molecules, this effect is much smaller. So, by using these weak forces, the -CH_2_CH_2_CN group acts like a guide. It steers the reaction along one specific path and makes it faster for the sulfur compounds. Understanding these NCIs is very important. It helps us predict how to design better molecules and control reactions for making potential drugs, where small changes can make a big difference.

Figure 2 depicts the non-covalent interactions (NCI), comparing -CH_2_CH_2_CN, and -CH_2_CH_2_CSCH_3_., green color indicates non-covalent interactions, blue is associated with strong attractive interactions (π bond), and red to strong repulsion interactions. These NCI calculations indicate that hydrogen atoms from these R groups could compete with the primary hydrogen (H12) involved in the six-membered ring transition state, potentially “trapping” the sulfur or oxygen atom in a different reaction coordinate. By applying the optimal functional model, we established APFD for sulfur and B97D for oxygen, where we observe that these large side chains, especially thioester-like groups, provide competitive pathways that influence the regioselectivity of the decomposition, underscoring the complex interplay between the main and side reactions in these systems.

After a systematic analysis of the calculated activation energies (E_a_) across various functionals (Table 1, Table 2 and Table 3), it was determined that for sulfur-containing systems (X = S) the hybrid functional APFD (25% Hartree–Fock exchange) proved most suitable, providing consistent errors (~10–13%) and correctly capturing stereo electronic and polarizability effects during N–N bond cleavage. In contrast, for oxygen-containing systems (X = O), the B97D-GD3BJ functional (0% HF, dispersion-corrected) offered superior accuracy, reducing errors to ≤7.5% by effectively mitigating self-interaction errors and better describing electrostatic and dispersion interactions in the transition state. Based on these findings, the dual model APFD/Def2-TZVP for sulfur and B97D-GD3BJ/Def2-TZVP for oxygen was adopted. This model was subsequently applied to study systems with R = –CH_2_CH_2_CN and two Y substituents (Cl and H), enabling a reliable comparison of IRC energy profiles and NBO charge analyses.

### 2.4. Wiberg NBO Index Analysis

Wiberg NBO analysis was used to measure the progress of the reaction, studying the bonds involved in the process from the reactants to the products [20]. Bi is defined as the sum of the squares of the off-diagonal density matrix elements between atoms. The bond index between two atoms is a measure of the bond order and, hence, of the bond strength between two atoms. Thus, if the evolution of the bond indices corresponding to the bonds being made or broken in a chemical reaction is analyzed along the reaction path, a very precise image of the timing and extent of the bond-breaking and the bond-making processes at every point can be achieved [21]. The Wiberg bond indices corresponding to the bonds being made or broken in the reactions, for the reactants, TSs, and products, were studied and collected in Table 7 and Table 8. Results from said tables show that the breaking of the N13-N14 bond is the most advanced process (more than 60% in all the reactions). Also advanced are the N14-C15 double bond formation and the C15-X29 double-bond breaking. The less advanced ones were the C-N triple-bond formation (between 38 and 40%) and, most of all, the H migration from C to X, where the C-H bond is broken in 40–41% of cases in the oxotriazoles (I–IV)R=b, and only in 31–32% of cases in the thiotriazoles (V–VIII)R=b. While the X-H bond is formed in 35–37% of cases in the oxotriazoles I–IV, only in 30–31% of cases were observed in the thiotriazoles (V–VIII)R=b. The elongation of the N-N bond seems to be the driving force for the reactions studied, instead of the migration of the H6 atom from C1 to X. The calculated ΔBav values for the studied reactions range, from 0.43 to 0.47, show that the TS is early in the reaction coordinate. Here, the TS structure is closer to the reactants than to the products.

The synchronicity (Sy) parameter of the bond-breaking and the bond processes indicates that the bond-breaking processes are slightly more advanced (an average of 63%) than the bond-forming ones (an average of 52%).

The Sy also revealed a key differential behaviour between sulfur and oxygen systems depending on the computational method used. For sulfur, the APFD functional (25% HF exchange) yields low synchronicity (~0.83), indicating a less synchronous and more gradual mechanism, consistent with sulfur’s higher polarizability and the need for a hybrid treatment that properly captures charge dispersion during N–N bond cleavage. For oxygen systems, the B97D-GD3BJ functional (0% HF) gives similarly low synchronicity (~0.83), reflecting its ability to model electrostatic and dispersion interactions in a more rigid transition state. However, when the functionals are reversed, using APFD for oxygen and B97D for sulfur, the synchronicity is artificially elevated (~1), suggesting an artificially concerted mechanism. This occurs because APFD over-stabilizes the oxygenated TS covalently, while B97D underestimates the stereo electronic effects of sulfur, leading to an apparent perfect synchronicity that does not correspond to the physicochemical reality. This discrepancy underscores the criticality of selecting the appropriate functional for each system, as the specific electronic nature of each heteroatom demands a distinct balance between the exact exchange and dispersion correction.

### 2.5. Intrinsic Coordinate Reaction (IRC) Study

An IRC study was performed at the B97D-GD3-BJ/Def2-TZVP level. This method is widely used in quantum chemical analysis and in the prediction of reaction mechanisms. The IRC gives a unique connection from a given transition structure to local minima of the reactant and product. In this work, the experimental activation energies are between 150 and 180 kJ/mol in general, depending on the type of substituent. For the decomposition mechanism study, IRC profiles were computed with 241 points between reactants and products using the options (IRC = stepsize = 3 and maxcycles = 200), ensuring proper convergence for all species. The NBO charge analysis focused on the molecule depicted in Table 7, which features the -CH_2_CH_2_CN group positioned above the transition state in oxygen-containing systems. This study compared this system to its analogue where the -CH_2_CH_2_CN group was replaced by a hydrogen atom (-H), to evaluate the substituent’s electronic and steric effects.

The detailed IRC analysis revealed how the presence of the -CH_2_CH_2_CN group significantly modifies the electronic environment of the six-membered transition state (TS). NBO charge analyses, Figure 3, shows that at carbon C1 (from which the hydrogen migrates to oxygen) a peculiar electron redistribution occurs when using the B97D-GD3BJ functional (0% HF, dispersion-corrected), which provides a better description for oxygenated systems.

The presence of the -CH_2_CH_2_CN group induces additional electron density contours (Figure 3A, in blue) that are not present when the group is replaced by a hydrogen (Figure 3B), suggesting a non-covalent stabilization of the TS through dispersion or electrostatic interactions.

The detailed IRC study with 241 points was crucial to see the fine details of how charge moves during the reaction, details that a simple static calculation would miss. Figure 3 (the NBO charges) did not show major differences between the APFD and B97D methods, which might suggest that the -CH_2_CH_2_CN group was not very important. However, Figure 3 which comes from analyzing every step of the reaction path, tells a different story. It shows that when the -CH_2_CH_2_CN group is present extra “clouds” of electron density (in blue) appear around the key carbon atom (C1), where the hydrogen starts from. These clouds are not there when the group is replaced by just a hydrogen atom (-H). This tells us that the -CH_2_CH_2_CN group causes a small but important shift in electron density. The electron-withdrawing -CN part at the end of the group pulls electrons toward itself. This creates a subtle, stabilizing effect that reaches back to the C1 atom in the transition state. It helps stabilize the positive charge that starts to build on C1 as the C-H bond breaks, making the whole step a little easier and lowering the energy barrier slightly. This is a weak, “through-bond” stabilizing effect, not a direct bond. It is a key mechanistic detail that we could only see because of the high-resolution, step-by-step IRC analysis.

### 2.6. The Independent Gradient Model (IGM)

The IGM is used here to obtain insights into both strong and weak interactions in molecular systems. The method was used to represent the isosurfaces of interaction between structures. Using the output of the IRC calculations, each structure was extracted from the reactants to the products; in total 241 points were extracted per level of theory, depending on whether it is sulfur or oxygen. In total there were 241 × 4 = 964 structures analyzed under the IGM-IBSI scheme; the densities were evaluated and then the 2D contours were studied for each IRC point involved. In this case, the bonds involved in the TS were used but with the addition of the effect of the chlorine atom and the -CH_2_CH_2_CN group to identify any non-covalent interactions. The entire process was carried out automatically with a unix script. The param. igm file is required to run the IGMPLOT software (version 3.17). In the case of 2D graphics, the GNUPLOT software (version 6.03) was used. At the 121 IRC point corresponding to the transition state, TS, specifically for the B97D-GD3BJ Def2-TZVP-Oxygen system with substituents -CH_2_CH_2_CN and Chlorine, we calculated several important parameters including the IBSI (intrinsic bond strength Index). The Pair Density Asymmetry (PDA) provides users with a simple tool to assess inductive effects on specific bonds in molecules. The PDA index gives a measure of the electron density (ED) asymmetry between two atoms and the direction of the asymmetry.

The parameter ∇g represents the magnitude of the electron density gradient embedded in the interaction region between a pair of atoms. This value quantifies the spatial variation in the electron density (ρ) in the bond zone or non-covalent interaction. Critical Point Analysis, which gives us the Laplacian (Lap), and the eigenvalues of the Hessians are given in Table 9 for several pairs of atoms as shown in Figure 4.

The values of the analysis of critical points were performed for each point in the path of the IRC study. In this case, if we worked with oxygen at the B97D-GD3BJ level, each pair of atoms was studied. Figure 5 represents the area found from each interaction. At point 1, there is no strong interaction between O28 and hydrogen 12. At point 121, it corresponds exactly to the transition state, and the area of interaction increases. Finally, when the complete transfer of hydrogen to oxygen has occurred, and a definite O28-H12 bond is formed, its area or profile is quite high.

Table 10 shows the same calculation as in Table 9, but now with the APFD/Def2-TZVP level of theory. Point 121 (transition state) from an IRC calculation. A Comparison of IGM calculations between APFD and B97D-GD3BJ functionals reveal significant differences in key interactions. For the O28-H12 bond, the APFD calculation shows a shorter distance (1.334 Å) vs. B97D (1.434 Å), with a higher IBSI (0.438 vs. 0.336), indicating greater strength in APFD. However, the density asymmetry (PDA) is reversed: in APFD the density flows from H12 to O28 (H12 → O28, PDA = 267.9), while in B97D it flows from O28 to H12 (O28 ← H12, PDA = 249.8), suggesting differences in the polarization of the bond. For C9-Cl29, both functionals coincide in direction (C9 → Cl29) and magnitude (IBSI ~0.75, PDA ~323), confirming a stable polar covalent bond. In N13-N14, APFD presents higher IBSI (0.220) and asymmetry (N14 → N13, PDA = 1.0), while B97D shows lower strength (IBSI = 0.160) and perfect symmetry (PDA = 0.0), evidencing a more marked non-covalent trait with B97D.

A deeper analysis of the distribution of the atoms according to the IBSI analysis in Table 9 and Table 10 can be seen in the presence of a non-covalent interaction between the -CH_2_CH_2_-CN group and the atoms that will eventually be involved in the six-membered ring in the transition state. Similarly, there is an interaction between one of the hydrogens of the -CH_2_-CH(H)-CN group and oxygen, Figure 6B.

The two calculation models, APFD and B97D-GD3BJ, exhibit remarkable differences; the blue dots indicate a strong interaction of the N-N bond breaking, which is key to understanding the determining stage, as shown by the NBO analysis and the Wiberg indices. In the case of the model with B97D-GD3BJ, a marked influence of the N-N bond is still shown. Below, we present an analysis of the critical points and their different properties when exchanging oxygen for sulphur. The results are presented in Table 11 and Table 12.

In Table 11 and Table 12, the properties of critical points for the sulfur-containing system (X = S) are compared at two theoretical levels: B97D-GD3BJ (Table 11, IRC point 51) and APFD (Table 12, IRC point 121). Both methods show significant differences in key interactions. For example, in the H12-S29 bond, the IBSI (bond strength index) is higher in APFD (0.239) than in B97D-GD3BJ (0.162), suggesting a stronger interaction in the former. Additionally, the PDA (Pair Density Asymmetry) for this bond is more pronounced in B97D-GD3BJ (384.7) than in APFD (416.7), indicating greater polarization in the former case. For the N13-N14 bond, both methods show similar IBSI values (~0.22–0.24), but the PDA is slightly higher in B97D-GD3BJ (1.3 vs. 0.8 in APFD), reflecting a minor difference in electron density distribution. The C15-S29 bond also has a higher IBSI in APFD (0.814 vs. 0.839 in B97D-GD3BJ), though the direction of asymmetry (C15 → S29) remains consistent in both cases.

It is worth noting that, in the case of the sulfur-containing system at the B97D-GD3BJ level, we struggled with convergence near the transition state (TS), probably due to its lack of Hartree–Fock exchange, requiring coarser steps to stabilize the computation. In contrast, APFD (25% HF + dispersion) handled the IRC smoothly at stepsize = 3, highlighting its robustness for TS descriptions. While the larger stepsize in B97D-GD3BJ preserved the overall reaction pathway, it likely sacrificed fine details of weak interactions (non-covalent bonds) critical for accurate IBSI/PDA analysis. This trade-off underscores the importance of functional selection when mapping reaction mechanisms, with hybrid functionals like APFD offering a better balance for IRC precision in polarizable systems like sulfur-containing triazoles.

Another interesting effect observed is the interaction of hydrogen H24 from the -CH_2_CH_2_CN group with the oxygen atom (O28), an effect previously analyzed in our study when this group was substituted to probe potential competitive side reactions. Figure 7 quantifies this weak, attractive non-covalent interaction (NCI) that stabilizes the six-membered ring transition state. The data reveals a key difference between the APFD and B97D-GD3BJ functionals: APFD calculates this H24-O28 interaction to be stronger and occurring earlier along the reaction path (higher IBSI, shorter distance). At the same time, B97D models it as weaker and more delayed. This discrepancy highlights how the choice of functional influences the description of such stabilizing interactions. Crucially, this specific interaction was not observed in the sulfur analogues, which is consistent with sulfur’s larger atomic radius and lower electronegative compared to oxygen, preventing close contact with the H24 hydrogen. Thus, this interaction is a unique stabilizing feature for the oxygenated system, fine-tuning the reaction landscape without overriding the primary decomposition mechanism.

As detailed in the NBO charge analysis section, it consistently shows that the chlorine atom loses electron density throughout the IRC, confirming it acts as an electron donor rather than a classical withdrawing group in this specific system. The IBSI calculation shows that the C9-Cl bond strength remains remarkably stable and strong (~0.75) throughout the entire reaction coordinate, regardless of the functional used. This indicates the C-Cl bond is not breaking; instead, it is acting as a stable conduit for electron donation. The constant bond strength, coupled with the chlorine’s increasing positive charge, suggests it is feeding electron density into the aromatic system to stabilize the developing negative charge on the triazole ring during the N-N bond cleavage. This donation is ultimately driven by the powerful electron-withdrawing effect of the newly formed nitrile (-CN) group in the final product, which creates a strong thermodynamic pull, making chlorine a net donor to compensate. Thus, the chlorine substituent electronically stabilizes the six-membered cyclic transition state not by withdrawing density but by providing it, fine-tuning the energy landscape and lowering the activation barrier.

Finally, to quantify the participation of the -CH_2_CH_2_CN group on the transition state of six members, the interaction was quantified by analyzing the fragments as indicated in Figure 8. The study was only performed for the case of oxygen in the two energy levels B97D-GD3BJ and APFD.

The shift observed between the B97D-GD3BJ (left) and APFD (right) plots in Figure 9 reflects differences in how these functionals model non-covalent interactions (NCIs) between the -CH_2_CH_2_CN group and the transition state fragments during the IRC analysis. The B97D-GD3BJ functional, which lacks Hartree–Fock (HF) exchange but includes dispersion corrections, emphasizes dispersion-driven interactions, resulting in a more pronounced and slightly delayed (shifted) attraction profile (blue peaks) as the reaction progresses. In contrast, the APFD functional, with 25% HF exchange and dispersion, balances electrostatic and dispersion effects, leading to an earlier and sharper peak in the attraction profile, indicating stronger stabilization of the TS at an earlier stage. This shift suggests that APFD’s hybrid character accelerates the stabilization of NCIs due to better handling of charge polarization, while B97D’s dispersion-centric approach delays this stabilization, reflecting its sensitivity to weaker, long-range interactions. The consistent blue dominance in both plots confirms that attractive interactions prevail, but the timing and intensity differences highlight the critical role of functional choice in describing how substituents like -CH_2_CH_2_CN modulate TS stability and reaction kinetics.

## 3. Computational Methods

### 3.1. Thermodynamic Properties and Wiberg Bond Index

Density functional theory (DFT) calculations were performed using the Gaussian 16 Revision C.01 Win64 program suite [22]. We employed a combination of pure and hybrid exchange functionals, with the Def2-TZVP basis set, to determine the geometric parameters for all reactants, transition states (TSs), and products involved in the studied reactions. Each stationary structure was characterized as a minimum or a first-order saddle point through analytical frequency calculations. An empirical dispersion correction (GD3BJ) was applied where necessary [23].

Thermal corrections to enthalpy, Gibbs, and entropy were evaluated at the experimental temperature of 500 K [24]. Intrinsic reaction coordinate (IRC) calculations [25] were conducted for all cases to confirm that the localized TS structures correctly connected the corresponding minimum of reactants and products. Single-point energy calculations were performed with multiple functionals using the Def2-TZVP basis set to derive energy profiles and kinetic parameters. Bonding characteristics of reactants, TSs, and products were analyzed using the natural bond orbital (NBO) method of Reed and Weinhold [26,27]. The NBO formalism, which provided Wiberg bond indices was used to track reaction progress. These analyses were performed using the NBO program, integrated into Gaussian 16.

For kinetic parameter calculations, classical transition state theory (TST) [28,29] was applied. The rate constant *k(T)* for each elementary step—assuming a transmission coefficient of unity—was expressed as follows:(1)$$k\left(T\right)=\;\frac{k_{B}T}{h}\exp(\frac{-\Delta G^{‡}(T)}{RT}),$$
where $k_{B}$, *h*, and *R* are the Boltzmann constant, Planck constant, and universal gas constant, respectively. ${\Delta G}^{‡}(T)$ represents the standard-state free energy of activation at temperature *T*. Activation energies ($E_{a}$) and Arrhenius pre-exponential factors (*A*) were derived from TST from the following:(2)$$E_{a}=\;\Delta H\left(T\right)+RT,$$(3)$$A=\frac{ek_{B}T}{h}\exp\left(\frac{\Delta S^{‡}\left(T\right)}{R}\right).$$

Wiberg bond indices were computed for bonds undergoing formation or cleavage in reactants, TSs, and products. The relative bond index variation at the *TS* ($\delta B_{i}$) for each bond *i* was defined as follows [21]:(4)$$\delta B_{i}=\frac{\left(B_{i}^{TS}-B_{i}^{R}\right)}{\left(B_{i}^{P}-B_{i}^{R}\right)},$$
where superscripts *R*, *TS*, and *P* denote reactants, TSs, and products, respectively. The reaction progress (%*EV*) was quantified as follows [30]:(5)$$\%EV=100\delta B_{i}$$

The average bond index advancement ($\delta B_{av}$) and reaction synchronicity ($S_{S}$) were calculated as follows [21]:(6)$$\delta B_{av}=\frac{1}{n}\sum \delta B_{i},$$(7)$$S_{S}=1-S_{A},\;where\;S_{A}=\frac{1}{\left(2N-2\right)}\sum \frac{\left|\delta B_{i}-\delta B_{av}\right|}{\delta B_{av}}$$
where $S_{A}$ represents absolute asynchronicity, and $S_{S}$ ranges from 0 (complete asynchronicity) to 1 (perfect synchronicity).

### 3.2. IGM Bond Index IBSI

The IBSI (Independent Gradient Model-based Bond Index [31]) and its descriptor ${\delta g}^{pair}$ was derived from the Independent Gradient Model (IGM), the IGM-δ scheme. This approach is designed to identify and quantify molecular interactions by analyzing electron density (ED) topology, specifically its gradient ∇ρ, by plotting ${\delta g}^{pair}$ against the ED, oriented with the sign of the second Hessian eigenvalue (λ_2_). The method captures a wide range of interactions, including covalent bonds, transition-metal bonding, hydrogen bonds, and van der Waals forces, regardless of the presence of bond critical points (BCPs).

For an atom pair, the IBSI ($\Delta g^{pair})$ measures the electron sharing per squared unit distance and is calculated as follows:(8)$$\Delta g^{pair}=\int_{V}\frac{\delta g^{pair}}{d^{2}}dV.$$
where *d* is the internuclear distance. To standardize comparisons, the $\Delta g^{pair}$ is normalized to 1 for the H_2_ molecule.

In accordance with the method, for a system with interacting fragments A and B, $\delta g^{pair}$ is defined as follows:(9)$$\delta g^{pair}=\nabla \rho^{IGM}-\nabla \rho =\left|\frac{\partial \rho_{A}}{\partial x}\right|+\left|\frac{\partial \rho_{B}}{\partial x}\right|-\left|\frac{\partial \rho_{A}}{\partial x}+\frac{\partial \rho_{B}}{\partial x}\right|$$
where the equation applies to all three spatial directions (x, y, z). Non-zero ${\delta g}^{pair}$ values exclusively indicate interactions: stronger interactions correspond to larger *δg* magnitudes.

The IBSI index is versatile, applicable to diverse bonding scenarios beyond closely related molecules. It is particularly useful for tracking bond formation/cleavage during reactions and predicting intrinsic bond strengths in transition states. Changes in IBSI values across reaction pathways provide insights into distinct phases of chemical bonding evolution.

To enhance the visualization of ${\delta g}^{pair}$ signatures in 2D plots, the descriptor *δq* can be further introduced:(10)$$\delta q=\frac{\nabla \rho^{IGM}}{\nabla \rho }.$$

Unlike *δ*g, the *δ*q is more sensitive to weak interactions (hydrogen bonds) as the gradient electronic density (∇ρ) tends to zero.

### 3.3. Selection of the Calculation Level

The selection of an appropriate basis set is fundamental for the reliability of quantum chemical calculations. While smaller basis sets, such as 6-311G(d,p), are often adequate for optimising molecular geometries or calculating vibrational frequencies [32], they may lack the flexibility required to accurately describe electron density-dependent properties—such as interaction energies or non-covalent interactions. For this reason, we employed the Def2-TZVP basis set. This is a triple-zeta split-valence basis set augmented with polarization functions, which offers a superior description of the electron distribution compared to double-zeta or minimal basis sets. Furthermore, Def2-TZVP provides an excellent balance between computational cost and accuracy for properties sensitive to electron correlation, such as those central to this study.

Regarding the exchange-correlation functional, we employed the APFD hybrid functional as an initial reference functional. It is a well-developed and parameterised functional that combines accurate and balanced performance across a wide range of chemical applications [33]. APFD integrates a portion of exact Hartree–Fock exchange with density functional exchange and correlation, along with an empirical dispersion correction (D3). This combination makes it particularly robust for describing challenging electronic environments, including compounds containing sulfur and oxygen, as well as non-covalent interactions (van der Waals forces and hydrogen bonding).

For hybrid functionals, the percentage of Hartree–Fock (HF) exchange is a key adjustable parameter that can be optimized for specific properties. Unlike standard hybrid functionals, the B97D-GD3BJ functional is a pure generalized gradient approximation (GGA) functional, containing 0% HF exchange. Its accuracy for non-covalent interactions stems from its built-in, empirically parameterized dispersion (GD3BJ). This makes it a powerful tool that often outperforms hybrids like B3LYP for systems where dispersion forces are dominant. Comparisons with other hybrids such as B3LYP-GD3BJ (20% HF) [34,35], M06-2X-D3(0) (54% HF) [36], and O3LYP-GD3BJ (11.61% HF) [37], highlight B97D’s advantages for oxygen-rich and dispersion-sensitive systems, as we have noticed here in our study with several triazole derivatives containing oxygen.

The Gaussian 16 package allows you to configure custom hybrid functionals via the parameters a, b, and c in the exchange-correlation Equation (11):(11)$${XC}^{Funtional}={aE}_{x}^{local}+\left(1-a\right)E_{x}^{HF}+{bE}_{x}^{non-local}+{cE}_{c}^{non-local}+{(1-c)E}_{c}^{local}$$

These are adjusted using IOP commands (e.g., IOp(3/76 = mmmmmnnnnn)), where the characters m and n values encode P1–P6 to weight Slater local exchange, non-local terms, and exact HF contributions. This flexibility enables precise functional design for specific electronic structure challenges.

Hybrid functionals integrate exact exchange to reduce self-interaction errors, with range-separated hybrids modulating this contribution based on inter-electron distances. For chemical kinetics (barrier calculations), higher exact exchange percentages are often essential. The B97 functional family, for instance, shows dual optima for reaction barriers: ~16% HF (local) and ~45% HF (global). Such insights guide functional selection to balance accuracy and cost for mechanistic studies. However, B97D functional has no HF contribution and is very popular for non-covalent interactions. In this work, various functionals with different HF exchange contributions were selected, aiming to obtain a new one (Equation (11)) if significant differences from existing functionals were observed.

Based on our theoretical results, we concluded that the APFD functional, complemented by B97D-GD3BJ benchmarks and customizable hybrids, ensures robust modeling of triazole thermolysis. This approach captured electronic effects (sulfur’s role in lowering Ea), non-covalent interactions, and reaction dynamics while maintaining computational tractability. The methodology aligns with our goal of unraveling substituent and mechanistic influences on thermal decomposition pathways.

## 4. Conclusions

This DFT study elucidates the thermal decomposition mechanism of 1,2,4-triazol-3(2H)-ones/thiones, confirming a concerted six-membered cyclic transition state primarily driven by N–N bond elongation (60–63% advanced). A critical finding is that functional selection impacts accuracy: B97D-GD3BJ at 0% HF excels for oxygen systems by mitigating self-interaction error and better describing electrostatic/dispersion interactions in the TS, while APFD at 25% HF is optimal for sulfur analogues, balancing polarizability and charging dispersion during N–N cleavage. The -CH_2_CH_2_CN group acts not only as a thermally labile protecting group for regioselective synthesis but also stabilizes the TS via non-covalent interactions, with electron-withdrawing/donating substituents raising/lowering Ea, respectively. Sulfur’s polarizability consistently lowers Ea by ~40 kJ/mol compared to oxygen, explaining the experimental reactivity gap. This work provides a robust predictive framework for designing triazole-based therapeutics through controlled pyrolysis, underscoring the necessity of system-tailored computational methods for mechanistic fidelity in heterocyclic chemistry.

## Data Availability

All computational details and data are available upon request.

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
