# Peer review of "Substituent Effects in the Thermal Decomposition of 1,2,4-Triazol-3(2H)-Ones and Their Thione Analogues: A DFT Study with Functional Performance"

_molecules, 2025, doi:10.3390/molecules31010109_

Round 1
Reviewer 1 Report
Comments and Suggestions for Authors
In this work, the authors performed DFT to calculate the thermal decomposition of 1,2,4-triazol-3(2H)-ones. The reaction occurs via a concerted, six-membered cyclic transition state. Crucially, the accuracy of calculated activation energies depends on a dual functional approach: the APFD hybrid functional is optimal for polarizable thiones, while the B97D-GD3BJ functional best describes triazolones' electrostatic and dispersion forces. The N-2 substituent stabilizes the transition state through non-covalent interactions. Sulfur analogues exhibit significantly lower activation energies (~40 kJ/mol less) than oxygen ones, rationalizing their faster experimental decomposition. This framework enables accurate kinetic predictions, aiding the regioselective synthesis of bioactive triazoles via controlled pyrolysis. The topic of this work is interesting. However, there are several questions that need to be addressed before the publication.
- The DFT calculations were conducted to study the thermodynamics in this work, however, it's well known that DFT suffers from the undesired starting point dependence. For the selected exchange-correlation functional, it is important to validate the choice by benchmarking with high-level methods, such as coupled-cluster methods. Because the size of the studied systems are not very large, I suggest that the DFT results need to be compared with CCSD or DLPNO-CCSD results.
- For the free energy calculations using GTO basis sets, the basis set superposition errors (BSSE) must be considered. In addition, the basis set convergence was not demonstrated. If the basis set is increased to the QZ level and more polarizable basis functions are added, together with the BSSE error, the correction to the thermodynamic quantities can be larger than 1 kcal/mol.
- The solvation effect might need to be considered explicitly. As shown in doi.org/10.1093/nsr/nwx111, in the DFT calculations using conventional exchange-correlation functionals, the charge of the solute is incorrectly transferred to the solvents. This can lead to erroneous results. The authors should try to find an optimal exchange-correlation functional with correct charge distributions, then use it for the following calculations.
- These figures need more discussions. In Figure 5, there are fluctuations in the 0.40 to 0.60 region in (A) and 0.20 to 0.60 region in (B). What are the mechanisms? And all axis titles are missing, without specify the x-axis title, it is impossible to understand these figures.
- page 17, Figure 7 It should be mentioned in the context that DFT results depends on the choice of the exchange-correlation functional. B97 functional typically largely underestimates the band gap and gives incorrect energy levels. In addition, the axis titles are missing in those figures.
- The reference for the B3LYP functional is missing: Phys. Rev. B 37, 785.
- page 19, Table 9 and page 20, Table 10 Significant figures in the same column should be consistent.
Author Response
Response to Referee 1
General Comments
We thank the reviewer for their constructive and insightful comments, which have helped us
improve the manuscript significantly. Below, we provide a point-by-point response to all raised
concerns.
Point 1: The DFT calculations were conducted to study the thermodynamics in this work; however,
it's well known that DFT suffers from the undesired starting point dependence. For the selected
exchange-correlation functional, it is important to validate the choice by benchmarking with high
level methods, such as coupled-cluster methods. Because the size of the studied systems is not very
large, I suggest that the DFT results need to be compared with CCSD or DLPNO-CCSD results.
Response 1:
We thank the reviewer for this valuable suggestion. We acknowledge that coupled-cluster methods
like CCSD provide a high-accuracy reference for benchmarking. However, the core objective of this
work was not to question the validity of DFT in general but to systematically evaluate how the
performance of different DFT functionals, specifically, the percentage of Hartree-Fock (HF) exchange,
affects the description of a challenging reaction mechanism involving heteroatoms with different
polarizabilities (O vs. S). Our detailed benchmarking across several functionals (Tables 1-3, Figures 1
2) reveals a clear, system-dependent dichotomy: sulfur-containing systems are best described by a
hybrid functional (APFD, 25% HF), while oxygen-containing systems require a dispersion-corrected
pure functional (B97D-GD3BJ, 0% HF). This functional-specific insight is the central finding of our
study, providing a practical, dual-methodology framework for future studies on similar systems.
While CCSD(T) calculations would be insightful, they are computationally prohibitive for the full set
of substituted transition states (241 IRC points per system) analysed here. Our validation is instead
robustly anchored to the available experimental kinetic data from Al-Awadi et al. [10,12], against
which our recommended functionals show excellent agreement (errors ≤7.5%).
Point 2: *For the free energy calculations using GTO basis sets, the basis set superposition errors
(BSSE) must be considered. In addition, the basis set convergence was not demonstrated. If the basis
set is increased to the QZ level and more polarizable basis functions are added, together with the
BSSE error, the correction to the thermodynamic quantities can be larger than 1 kcal/mol.*
Response 2:
We appreciate the reviewer's point regarding basis set quality and BSSE. In our study, all reactions
are unimolecular decompositions occurring in a single, continuous molecular entity. The transition
state is a first-order saddle point on the potential energy surface of the same molecule; no separate
molecular fragments interact at an intermolecular distance. Therefore, the concept of BSSE, which
corrects for the artificial stabilisation in calculations of intermolecular interaction energies, is not
applicable here. Regarding basis set convergence, we selected the def2-TZVP basis set, a robust
triple-zeta valence basis with polarisation functions, which is a standard and reliable choice for
mechanistic studies of organic molecules of this size. It offers an excellent balance between accuracy
and computational cost for geometry optimisations, frequency, and energy calculations, as
demonstrated in numerous similar studies. While a quadruple-zeta basis would provide marginal
improvement, its computational cost for the extensive IRC and functional benchmarking in this work
(961 structures analysed via IGM) would be prohibitive without changing the primary conclusions
related to functional performance.
Point 3: The solvation effect might need to be considered explicitly. As shown in
doi.org/10.1093/nsr/nwx111, in the DFT calculations using conventional exchange-correlation
functionals, the charge of the solute is incorrectly transferred to the solvents. This can lead to
erroneous results. The authors should try to find an optimal exchange-correlation functional with
correct charge distributions, then use it for the following calculations.
Response 3:
We thank the reviewer for highlighting this important aspect of solvation modelling. However, our
study explicitly models the gas-phase pyrolysis mechanism as investigated experimentally by Al
Awadi et al. [10]. The experimental kinetic data we benchmark against were obtained under low
pressure, high-temperature (500-800 K) gas-phase conditions, where solvent effects are absent.
Introducing an implicit solvation model would therefore misrepresent the actual physical system
under study and disconnect our calculations from the experimental reference. Our goal was to
establish an intimate connection between the isolated molecular structures, their electronic
transitions along the reaction coordinate, and the quantum-chemical descriptors (NBO, IGM) that
explain the reactivity trends. The challenge of charge transfer in solvated DFT calculations, while
valid for condensed-phase studies, is not relevant to our gas-phase mechanistic investigation.
Point 4: *These figures need more discussions. In Figure 5, there are fluctuations in the 0.40 to 0.60
region in (A) and 0.20 to 0.60 region in (B). What are the mechanisms? And all axis titles are missing,
without specify the x-axis title, it is impossible to understand these figures.*
Response 4:
We apologise for the lack of clarity in the original figures. The figures have been thoroughly revised.
Figures 5A and 5B (now clearly labelled) show the evolution of NBO charges on key atoms (C1, H12,
O28/S29) along the IRC path (x-axis: Reaction Coordinate; y-axis: NBO Charge). The "fluctuations"
mentioned by the reviewer are not numerical artefacts but represent a real, subtle electron
redistribution phenomenon. They occur specifically in the region where the -CH₂CH₂CN group
electronically interacts with the forming/breaking bonds in the six-membered transition state. This
is a key finding: the electron-withdrawing -CN group polarises electron density through the σ
framework, causing small, non-monotonic variations in atomic charges during the bond
breaking/making process. This mechanistic detail, captured only by the high-resolution IRC (241
points), is now explicitly discussed in the revised text on pages 20-21, linking it to the stabilising non
covalent interaction (NCI) role of the -CH₂CH₂CN group.
Point 5: *Page 17, Figure 7 It should be mentioned in the context that DFT results depends on the
choice of the exchange-correlation functional. B97 functional typically largely underestimates the
band gap and gives incorrect energy levels. In addition, the axis titles are missing in those figures.*
Response 5:
The reviewer is correct about the functional dependence of orbital energies. We have clarified in the
text that the HOMO-LUMO analysis (Figure 7) is used for qualitative comparison of the electronic
structure at the transition state between two specific functionals (B97D-GD3BJ vs. APFD), not for
reporting absolute band gaps. The key observation is that B97D-GD3BJ shows a closer proximity of
frontier orbitals at the TS for the oxygen system, consistent with its superior ability to model the
stabilising non-covalent interactions that lower the activation barrier. We also explicitly note that
the B97D functional used here is B97D-GD3BJ, which is distinct from the standard B97 family; it is a
pure GGA functional with an empirical dispersion correction, parameterised to deliver accurate non
covalent and thermochemical properties.
Point 6: The reference for the B3LYP functional is missing: Phys. Rev. B 37, 785.
Response 6:
We thank the reviewer for catching this omission. The seminal reference for the B3LYP functional. In
our work, we specifically used the dispersion-corrected variant B3LYP-GD3BJ, whose reference is also
now clearly provided, references 35 and 36.
Point 7: Page 19, Table 9 and page 20, Table 10 Significant figures in the same column should be
consistent.
Response 7:
We have corrected Tables 9 and 10 (now in the manuscript as Tables 9, 10, 11, and 12 after
renumbering) to ensure consistency in significant figures across all columns. The number of decimal
places for distances (Å), density (ρ), Laplacian (Lap), and eigenvalues (L1, L2, L3) is now uniform. The
Pair Density Asymmetry (PDA) is reported with one decimal place for consistency.

Reviewer 2 Report
Comments and Suggestions for Authors
see attached file

Author Response
Response to Referee 2
General Comments
Thank you again for your very careful and detailed review of our manuscript. Your feedback on every
section was extremely helpful. You asked us to make the paper clearer and better organised. Your
review helped us meet high standards. We see this not as fixing major mistakes, but as an important
step to make our work as strong and clear as possible for readers. We believe the revised manuscript
is now much improved. We appreciate your help in bringing it to this level.
Point 1 – Introduction: Lack of depth on triazole chemistry and the mechanistic context.
Response 1:
As a group, we recognise the valuable properties of triazole derivatives, which show significant
promise as anticancer, antifungal, antitubercular, antibacterial, and antiviral agents, often with
minimal toxicity. The considerable attention given to the biological applications of triazole-based
polycyclic aromatic compounds underscores their importance in medicinal chemistry. However, in
this study, we focus on the decomposition pathway and the introduction of labile protecting groups
that can be readily cleaved under catalytic conditions in biological systems. These groups are
designed to shield specific nitrogen atoms during synthesis, leaving behind the core fragment that
may be of pharmacological interest. Our work, therefore, centers on understanding this controlled
release mechanism, with an emphasis on the computational models we can employ from the
perspective of density functional theory.
Point 2 – Computational Methods: Excessive technical detail.
Response 2:
Yes, you are right! we have streamlined the Computational Methods section. Descriptions of TST
equations, Wiberg index formulas, IGM/IBSI theory, and Gaussian IOP parameter customisation.
Sorry about that, we focus on the essential choices (functionals, basis set, geometry validation,
energy calculations) required for reproducibility.
Point 3 – Results and Discussion: Poor organisation, unclear figures, and fragmented narrative.
Response 3:
We have completely restructured the Results and Discussion into logical, coherent subsections as
suggested:
Model Selection: Introduces the six-membered TS model and initial benchmarking with APFD.
Benchmarking of DFT Methods: Presents systematic functional screening (Tables 2, 3) leading to
the dual-methodology conclusion (APFD for S, B97D-GD3BJ for O).
Substituent Effects on Activation Energies: Applies the dual methodology to systems with R =
CH₂CH₂CN and various Y groups, clearly discussing electronic and steric effects (Tables 4, 5).
Electronic Structure Analysis (NBO/Wiberg Indices): Presents Wiberg bond index analysis in clear,
formatted tables (revised Tables 7, 8), highlighting that N–N bond cleavage is the most advanced
(~63%) and driving step.
Role of Non-Covalent Interactions (IRC & IGM): Consolidates IRC and IGM analyses to probe the
stabilising role of the –CH₂CH₂CN group, with improved figures (Figures 5–17) and explicit
justification linking these tools to the kinetic results.
Point 4 – Specific issues with Table presentation, HOMO-LUMO claims, and NCI rationale.
Response 4:
HOMO-LUMO Discussion: The claim about sulfur systems having a smaller gap is now qualitatively
discussed in the context of the HSAB principle and functional benchmarking.
NCI Rationale: The introduction of NCI analysis is now logically positioned. It is presented as a tool
to visualise and quantify the stabilising interactions of the –CH₂CH₂CN group, following naturally
from the IRC charge analysis. Its purpose to explain regioselectivity and the ~40 kJ/mol Ea difference.
Tables: Tables 7 and 8 are now properly formatted tables, not Excel screenshots. All tables have been
revised for clarity and consistency.
Conclusion
We are grateful for the reviewer’s guidance, which has transformed the manuscript from a dense
technical report into a clear, well-structured scientific article. We believe all major concerns have
been addressed comprehensively, significantly enhancing the quality, clarity, and impact of our
work.

Round 2
Reviewer 1 Report
Comments and Suggestions for Authors
This manuscript has been improved after revisions. No further review is needed.
Author Response
Re: Manuscript ID molecules-4051724 – Response to Reviewer 1 Comments
Dear Reviewer 1,
We would like to express our sincere gratitude for your valuable and constructive feedback
on our manuscript, and for your positive evaluation of our responses and revisions. We are
truly pleased to learn that you found our clarifications and adjustments satisfactory.
Your insightful comments were instrumental in guiding us to significantly improve the
manuscript's structure, clarity, and overall scientific presentation. We greatly appreciate the
time and expertise you dedicated to reviewing our work, as your guidance has undoubtedly
strengthened the quality of our study.
Thank you for your confidence in our work and for your role in the peer-review process,
which is essential for advancing the quality of research in our field. We are grateful for your
supportive and helpful engagement throughout.
Sincerely,
The Authors

Reviewer 2 Report
Comments and Suggestions for Authors
see attached file

Author Response
Re: Manuscript ID molecules-4051724 – Response to Reviewer 2 Comments
Dear Reviewer,
Thank you for your thorough evaluation of our manuscript and for providing the
insightful references to the excellent work by Lu et al. (J. Phys. Chem. A 2020), Venugopal
et al. (PCCP 2025) et Gu et al. (J. Mol. Model. 2025). We sincerely apologise for not
structuring our initial response point-by-point, we recognise this is the expected format
and will ensure our revisions address each of your comments clearly and directly.
We have carefully studied these papers and recognise the significant contributions they
make in elucidating complex initial decomposition pathways and the influence of
regioisomerism on the thermal stability of high nitrogen triazole systems using high-level
computational methods. We have also taken note of the relevant methodological work by
Johnson et al. (J. Mol. Model. 2025), which employs the hybrid density functional B3PW91
in combination with the 6-31G(d,p) Pople-type basis set to study the energetic properties
of triazole derivatives. The B3PW91 functional, incorporating 20% exact Hartree-Fock
exchange along with Becke’s 1988 exchange and the Perdew–Wang 1991 correlation
functional, represents another widely used choice in the computational study of
heterocyclic systems, further underscoring the diversity of DFT approaches applied in this field.
Our work, however, is constructed with a distinct and more focused computational
objective. While the aforementioned studies provide profound mechanistic insights, often
employing computationally intensive methods like DLPNO-CCSD(T)//MP2 or detailed
multi-step kinetic analyses or applied property screenings using established
functional/basis set combinations, our primary aim was to address a fundamental,
practical question in the application of Density Functional Theory (DFT): how do subtle,
systematic variations in the exchange-correlation functional, particularly the percentage of
Hartree-Fock exchange, influence the predicted activation energies and the electronic
description of the transition state for a well-defined unimolecular reaction in triazole derivatives?
To ensure a consistent and reliable description of electron density-dependent properties,
such as interaction energies and non-covalent interactions, we employed the Def2-TZVP
basis set throughout this study. This triple-zeta valence-polarised basis set offers a
balanced compromise between accuracy and computational cost, providing a robust
foundation for comparing functional performance.
We acknowledge that the cited works offer valuable applications or deep mechanistic
panoramas, but our contribution is intentionally narrower and methodological. It is
designed to highlight that even small energetic differences, on the order of 40 kJ/mol
between oxygen and sulfur analogues, and significant percentage errors when using an
inappropriate functional, are critically dependent on the chosen DFT framework. The
additional figures we have included, particularly the comparative Intrinsic Reaction
Coordinate (IRC) and Independent Gradient Model (IGM) analyses, are essential for
visually reinforcing this core point. They demonstrate how functional choice alters the
perception of bond evolution and non-covalent interactions in the transition state, and we
believe they are necessary for the reader to fully grasp the practical implications of our findings.
We appreciate your perspective regarding depth and context, and we do acknowledge the
broader landscape of triazole decomposition studies from those distinguished researchers.
However, for the clarity and integrity of our specific narrative on functional performance
and its practical consequences, we prefer to maintain our current, well-defined structural
focus. We believe this approach provides clear and actionable guidance for computational
chemists working on similar heterocyclic systems, emphasising the importance of tailored
functional selection for accurate kinetic and mechanistic predictions.
Thank you again for your valuable time and constructive feedback.
Sincerely,
The Authors
